# Urban Areas Create Refugia for Odonates in a Semi-Arid Region

**DOI:** 10.3390/insects12050431

**Published:** 2021-05-11

**Authors:** Danielle M. Husband, Nancy E. McIntyre

**Affiliations:** Department of Biological Sciences, Texas Tech University, Lubbock, TX 79409-3131, USA; danielle.husband@ttu.edu

**Keywords:** land use, Odonata, playa wetland, Texas

## Abstract

**Simple Summary:**

In semi-arid regions like western Texas (USA), there is limited natural habitat available for wetland organisms like odonates (dragonflies and damselflies), due to water scarcity that is compounded by anthropogenic land-use activities (primarily agriculture) that compromise water presence and quality. Other forms of anthropogenic land use, however, create wetland habitat for regional biodiversity through the construction of urban stormwater catchments. We surveyed adult odonates at 133 wetlands (49 in natural grassland settings, 56 in cropland, and 28 in urban areas) in western Texas from 2003–2020. Playas in an urban setting had greater species richness than those surrounded by grassland or cropland. We recorded 33 odonate species, seven of which were found only in urban wetlands, compared to two unique species in cropland wetlands and one unique species in grassland wetlands. The remaining 23 species occurred in multiple wetland types. The odonate community in urban wetlands was distinctly different from those in non-urban wetlands. Urban wetlands were not larger in surface area than the other wetland types, but because they were fed from more consistently available urban runoff, they held water longer, even during severe regional droughts. By concentrating water in an otherwise dry area, human environments can support more odonate species than would otherwise be present. Thus, although anthropogenic activities are often seen as being detrimental to biodiversity, some activities can actually create refugia for wildlife.

**Abstract:**

In western Texas, most wetlands are fed from precipitation runoff, making them sensitive to drought regimes, anthropogenic land-use activities in their surrounding watersheds, and the interactive effect between these two factors. We surveyed adult odonates in 133 wetlands (49 in grassland settings, 56 in cropland, and 28 in urban areas) in western Texas from 2003–2020; 33 species were recorded. Most species were widespread generalists, but urban wetlands had the highest species richness, as well as the most unique species of any of the three wetland types. Non-metric, multidimensional scaling ordination revealed that the odonate community in urban wetlands was distinctly different in composition than the odonates in non-urban wetlands. Urban wetlands were smaller in surface area than the other wetland types, but because they were fed from more consistently available urban runoff rather than seasonal precipitation, they had longer hydroperiods, particularly during a multi-year drought when wetlands in other land-cover contexts were dry. This anthropogenically enhanced water supply was associated with higher odonate richness despite presumably impaired water quality, indicating that consistent and prolonged presence of water in this semi-arid region was more important than the presence of native land cover within which the wetland existed. Compared to wetlands in the regional grassland landscape matrix, wetlands in agricultural and urban areas differed in hydroperiod, and presumably also in water quality; these effects translated to differences in the regional odonate assemblage by surrounding land-use type, with the highest richness at urban playas. Odonates in human environments may thus benefit through the creation of a more reliably available wetland habitat in an otherwise dry region.

## 1. Introduction

An estimated 33% of wetlands globally have been lost to anthropogenic activities, primarily land use/land cover changes [1]. Such losses can be most acutely detected in arid and semi-arid regions with seasonal rains, where water is inherently scarce and drought is common [2,3,4,5]. Although anthropogenic activities that alter water availability or quality are associated with a decline of biodiversity [1,2,3,6], other activities can increase the availability of water in the landscape, with a potentially positive effect on regional biodiversity. For example, natural wetlands that have been modified to retain stormwater are widely recognized as supporting odonate diversity by providing aquatic resources despite differences in urban water quality relative to wetlands in unmodified landscapes [7,8,9,10]. Thus, although land use/land cover change is ultimately responsible for most wetland losses, it can also augment the availability of water, an effect that is particularly important in otherwise dry regions [9,10].

The Great Plains of North America is one such region where these conflicting aspects of anthropogenic activities on water availability are apparent. Over 88% of the grasslands of the Great Plains have been converted to tilled cropland and pastures for livestock grazing; ~<1% of the area is urbanized [11]. This area is also characterized by ~80,000 temporary, freshwater wetlands known as playas [12] (Figure 1). As the primary source of aboveground freshwater in an otherwise semi-arid region, playas are foci for biodiversity [13,14]. These wetlands are primarily fed via surface runoff from seasonal (June–September) precipitation [15] and are dry more often than they are wet [16,17]. As depressional wetlands, playas are highly influenced by land use/land cover in their surrounding watersheds and precipitation availability, which jointly affect the amount of runoff reaching a playa [18,19,20]. Playas in agricultural settings experience inputs of agrochemicals and erosional sediments in runoff [12,14,21,22], and urbanization likewise affects water quality via pollutants [9,23].

The portion of the Great Plains with the highest density of playas is in the Texas panhandle [24] (Figure 1). Over 20,000 playas occur within this region, with ~99.8% directly affected by human modifications, such as pumps, livestock watering structures, cultivation within the wetland basin, and others [12,15]. This region is characterized by grasslands that are largely used for the grazing (~49% land area) and agriculture of annual plants, primarily cotton, corn, sorghum, and wheat (~46% land area under cultivation) [25]. The cities of Lubbock and Amarillo, as well as numerous smaller towns, represent only a small proportion of the regional land area, but contain playas that hold water during otherwise dry periods because of more consistent availability of urban runoff from anthropogenic sources [20,26,27,28] (Figure 1). This region thus provides an exemplar of how land use/land cover activities influence the presence of water, with consequences for wetland-dependent wildlife.

Odonates (Odonata: dragonflies and damselflies) are one such group that should be affected by anthropogenic influences on playas of the Great Plains. These semi-aquatic invertebrates are reliant on freshwater areas for reproduction and survival, with different species requiring different hydroperiod lengths for maturation. Odonate maturation varies from weeks to years, depending on a range of abiotic factors at their natal geographic location, including seasonal temperature fluctuations, rainfall, and subsequent food resource availability [29]. In general, smaller-bodied damselflies (suborder Zygoptera) have shorter development times than do larger-bodied dragonflies (suborder Anisoptera), but there are differences even within each suborder. For example, the dragonfly *Pantala hymenaea* can complete its nymphal development in as few as five weeks, comparable to some damselflies [29]. Even migratory species that can find widely scattered wetlands may not breed there successfully if those wetlands have short hydroperiods. Playas are dry more frequently than they are wet, and indeed can be dry for years at a time [16]. The ephemeral nature of playas should thus preclude the presence of species with protracted development times in much of the Great Plains.

However, just as in other parts of the world, some anthropogenic activities can increase the hydroperiod in the wetlands of this region by deepening basins or providing supplemental water, such as from irrigation runoff. For example, in a study of 8404 playas from 2008 to 2011 in Texas, only urban playas (*n* = 25) held water the entire time, whereas 3726 playas surrounded by cropland never held water during that four-year span, and the remaining ones (consisting of those surrounded primarily by cropland or grassland) were intermittently wet and dry [20]. Moreover, urban playas were the primary source of water during a long-term, exceptional drought in this region [22]. Playas in Texas have hydroperiods ranging from 18 to 453 days, depending on land use: playas surrounded by grassland were wet nearly twice as long as those surrounded by crop fields [30,31]. In another study, the hydroperiod was strongly influenced by rainfall and surrounding land use, with a median hydroperiod of 109 days but a range of 16 to 1312 days [20]. Although some previous studies [18,19] have found that playas surrounded by cropland were inundated more frequently than those surrounded by grassland, there were differences between different forms of grassland (including grazed rangelands and former croplands restored to grassland under the U.S. Department of Agriculture’s Conservation Reserve Program). Other work, however, found that wet playas in Texas were more likely to be in grassland than cropland [20]. Thus, playa hydroperiods vary greatly by land use/land cover context and may support different local odonate communities.

There are currently 324 odonate species in the Great Plains, with 104 that have been documented in the Texas panhandle (based on data from OdonataCentral: https://www.odonatacentral.org/, accessed on 24 January 2021). Previous studies have determined that odonate species richness at the playas of the Texas panhandle is influenced by the presence of cropland or grassland in the surrounding watershed [32,33]. However, the importance of urban wetlands on odonates in this area is still largely speculative. Given that urban playas tend to have prolonged hydroperiods relative to playas in other land use/land cover contexts [20], they may support a different odonate assemblage, characterized by species with longer ontogenetic development periods that are accommodated by longer hydroperiods. There should thus be differences in the odonate assemblage at wetlands based on their land use/land cover-driven hydroperiod, but this has not yet been examined, particularly with respect to a potentially positive effect that urbanization may play.

We compared odonate occurrence at playas surrounded by the three most prevalent land use/land cover types in this region (grassland, cropland, urban). We hypothesized that urban playas would have the highest odonate richness, given their extended water availability. In addition, because dragonflies are generally larger-bodied than are damselflies, with longer developmental periods that require longer hydroperiods [29], we expect dragonfly richness to be greater at urban playas than at non-urban ones. Although land use/land cover also affects playa water quality [22,31,34,35,36], the effects of water quality on odonates of playas are unknown in this region. Because odonates are not as sensitive to water quality as aquatic invertebrates from the orders Ephemeroptera, Plecoptera, and Trichoptera [37], our focus was on how odonates respond to human-dominated environments through the effects of the availability of water as a function of land use/land cover. Gaining such information is a necessary first step for more focused, in-depth studies (e.g., on the effects of land use/land cover on water quality and hence on odonates of this region), and subsequently, for invertebrate conservation and water management recommendations.

## 2. Materials and Methods

### 2.1. Study System 

In summer 2020, D.M. Husband surveyed 24 urban playas in the cities of Brownfield, Levelland, Lubbock, Muleshoe, and Plainview for odonate occurrence and richness (Figure 1). Urban wetlands are known colloquially in this region as “playa lakes” because of their extended hydroperiods. In addition to this dataset, data from non-urban playas (*n* = 105; 49 grassland, 56 cropland) and urban playa lakes (*n* = 21) from N.E. McIntyre, spanning 2003–2019 (hereafter, McIntyre Lab Data) were also used to create a comprehensive evaluation of playa odonates as a function of surrounding land use/land cover in the Texas panhandle.

### 2.2. Data and Collection Methods 

Adult odonates were surveyed systematically at each playa by walking in the littoral zone, with survey lengths and durations commensurate with playa surface area. Surveys were conducted from May–September on days above 23 °C, with clear to moderately clear skies and low wind. We recorded the presence of each species found at each playa, with species identifications confirmed via photographs or collection of vouchers; specimens are housed in the Department of Biological Sciences at Texas Tech University.

We used occurrences rather than counts of individuals per species, because abundances of adult odonates can be very difficult to determine accurately, being dependent on species conspicuousness as a function of sex, territoriality, body size, and weather conditions [29,38,39]. Therefore, we instead used species frequency of occurrence across playas as a proxy for abundance counts. Our surrogate of abundance (commonness) was scaled by the frequency of occurrence (i.e., if species A was sighted at 16 playas, it was given an occurrence rating of 16, and this would be greater than a species sighted only once (occurrence rating of 1)). Additionally, we focused on adults, because identifying exuviae or nymphs (particularly of early instars and of Zygoptera) of species in this area is difficult, due to a lack of taxonomic keys (at least for Zygoptera, and even keys for Anisoptera are for F-0 nymphs and not earlier instars). Because adults fly, it is possible that their presence at a given playa may not accurately indicate whether they had emerged from that playa or could use that playa for successful reproduction; however, it does indicate their occurrence and present use [38,40].

Some playas were visited on multiple dates, whereas others were visited only once. Sampling visits were conducted regardless of the presence of water in the playa basin. At each visit, surrounding land use/land cover within 200 m of each playa was assessed visually, and playas were classified as urban, grassland or cropland. The presence of water was noted during each visit; these occurrences were tallied, and the frequency of visits when water was detected was calculated by land-use type.

### 2.3. Odonate Assemblage Characterization and Analysis

Species occurrences were tallied across wetland types (urban, grassland, or cropland). Because there were different numbers of playas belonging to each of the three land-use types, we built species accumulation curves using the *vegan* package in RStudio 1.4.1103, to ensure that comparisons of species richness would not be a function of sample size [41,42]. Species accumulation curves were used to determine whether the number of playas sampled had been adequate to represent richness at each of the three land-use types, and to compare estimated richness across the land-use types.

Using species presence/absence frequency data by site, a non-metric, multidimensional scaling (NMDS) ordination was conducted with the metaMDS function from *vegan* in RStudio 1.4.1103 [41]. NMDS is one of the most flexible forms of analysis of complex biodiversity datasets, capable of handling data that are not normally distributed and not requiring that there be linear relationships among variables [43]. We used the Bray–Curtis dissimilarity metric and stress plots to find the optimal number of axes that reduced stress without the possibility of overfitting (*k* = 3) [41]. The ordination axes were then used as dependent variables to examine the influence of land-use type on the structuring of species across sites. An analysis of variance (ANOVA) was performed to determine if odonate communities differed across land use/land cover types.

## 3. Results

Thirty-three odonate species were detected across all playas (Table 1). Urban playas (*n* = 28) had the highest overall richness at 30 species, compared to 23 at grassland playas (*n* = 49) and 25 at cropland playas (*n* = 56) (Figure 2). This is notable because grassland playas were sampled the most (869 sampling events compared to 477 at cropland playas and 48 at urban playa lakes; Figure 3 and Figure 4). Twenty-one species were found at all three playa types (Figure 2). Seven species were solely found at urban playas. Cropland and grassland playas shared no species that were not also found at urban playas, grassland and urban playas shared one species not found at cropland playas, and urban and cropland playas shared one species not found at grassland playas (Figure 2).

Despite this overlap, there were enough uniquely occurring species to be able to distinguish the odonate community at urban playas from those at non-urban ones. The odonate communities at grassland and cropland playas were nearly identical in terms of species presence/absence, as seen by a strong overlap in the 95% confidence ellipses of the centroids of the distributions of species across sites by the three land use/land cover types within the ordination space (NMDS; Figure 5). This result was confirmed via ANOVA (*F*_2,126_ = 325.95; *p* < 0.0001), and differences in the first NMDS ordination axis by land use/land cover types were corroborated with Tukey’s honestly significant difference post-hoc tests. Cropland and grassland playas were not significantly different from each other in odonate composition (*p* = 0.96), but each differed significantly from urban playas (grassland: *p* < 0.0001; cropland: *p* < 0.0001). Seven species were found only at urban playas (*Argia apicalis*, *Brachymesia gravida, Enallagma basidens, Ischnura barberi*, *I. posita*, *I. ramburii*, *Telebasis salva*), two only at cropland playas (*Celithemis eponina* and *Dythemis fugax*), and one only at grassland playas (*Erythrodiplax umbrata*) (Figure 6). Only one species was found at both cropland and urban playas (*I. damula*). One species was found at both urban and grassland playas (*Erythemis vesiculosa*). Urban playas had the greatest number of damselfly species, recorded at 13 species, compared to 6 at grassland playas and 7 at cropland playas. *Ischnura* was the most diverse genus, with seven species recorded. No odonate species were observed at one urban playa and three grassland playas (one surveying visit apiece). Only one species, *T salva*, was detected only once in this study (found at an urban site).

Estimated species richness at both cropland (*n* = 49) and grassland (*n* = 56) playas reached an asymptote at around 30 playas (Figure 7), indicated that surveying ~30 playas was sufficient for representing adult odonate richness in these two land-use types. The lack of such an asymptote for urban playas, however, indicates that there likely are still some odonate species that are regionally present but which have not yet been documented at urban sites (Figure 7). Thus, even though the most species were found at urban playas, if the number of sampling visits were comparable to the sampling that has been conducted at grassland and cropland playas, even more species would likely be encountered.

Additionally, urban playas had the highest odonate richness, even though they were smaller in average surface area (0.09 km^2^) than grassland (0.15 km^2^) or cropland (0.11 km^2^) playas. This finding is particularly notable, because the length and duration of surveys were proportional to playa surface area (see Section 2, Materials and Methods), so the greater species richness at urban playas was not a function of their size (actually, the result was the contrary).

Of the playas surveyed from 2009–2020, urban playas held water ~98% of the time, whereas grassland and cropland playas held water about half the time (Table 2). Because sampling visits occurred regardless of the presence of water, these results indicate that the presence of water was associated with human environments.

## 4. Discussion

In examining odonate occurrence at playas surrounded by the three most prevalent regional land use/land cover types (grassland, cropland, urban), we found that playas in an urban setting had greater species richness than those in grassland or cropland. Moreover, urban playas supported the greatest number of unique species not found at other land use/land cover types (*n* = 7; Figure 3); these playas were more reliable sources of water when other playas were dry, a finding that is commensurate with studies from other regions that have documented greater [8] or comparable [7,9,44] odonate species richness in human-made wetlands compared to “natural” ones. In addition, this finding supports our initial ideas about the importance of water availability in arid and semi-arid regions, particularly with respect to the increased hydroperiod length of urban playas compared to ones in cropland or grassland. However, just as urban ponds elsewhere in the world tend to feature widespread generalist odonate species [10,45], most of the species in our study (*n* = 21) were generalists that occurred in all three playa types (Figure 2). We documented seven species that were found only in urban wetlands: *Argia apicalis, Brachymesia gravida, Enallagma basidens, Ischnura barberi, I. posita, I. ramburii*, and *Telebasis salva*. In comparison, only two species were found only at cropland playas (*Celithemis eponina* and *Dythemis fugax*) and one only at grassland playas (*Erythrodiplax umbrata*). As such, the odonate assemblage present at urban wetlands differed from those at non-urban wetlands, illustrated by the separation of the NMDS 95% confidence ellipses between urban and the remaining playa land use/land cover types; grassland and cropland playas had significant overlap and similar species detected (Figure 5).

Our focus on adult odonates and our use of incidence as a proxy of abundance (in lieu of counts of individuals per species) may have caused us to underestimate responses to specific land use/land cover types. Occurrence is positively associated with abundance, though it should be relatively less sensitive in detecting temporal population trends [46], which were not a focus in our study. There are also drawbacks in using presence of adults (the vagile life stage) to represent populations [38,47]. Surveys that focus on odonate adults are far more common than those that include nymphs or exuviae [48,49]. With adults, however, it is possible that some species at a playa may have dispersed there but would be unable to reproduce there because of too-short hydroperiods. Most odonate species studied to date do not disperse more than a few hundred meters from their natal site, although a few are capable of long-distance migration [50]. Thus, the question remains: were the species we detected as present at a site ones that had successfully emerged from that playa, or were they vagrants/transients just passing through?

The urban, cropland, and grassland playas had equal numbers of dragonfly species (*n* = 17 each), which does not support our notion that the longer hydroperiods of urban playas would support more dragonflies, which on average have longer aquatic (nymphal) development times than do damselflies. Rather surprisingly, urban playas supported the greatest number of damselfly species (*n* = 13). Damselflies, being smaller-bodied, have shorter development times, indicating that their greater measured prevalence at urban playas was due to factors other than duration of water availability, such as habitat structure or prey availability. Although damselfly abundance may be reduced at urban wetlands [51], our findings support those of Perron et al. [9], who found greater damselfly diversity at urban stormwater ponds than at natural areas. This group of insects may be more tolerant of anthropogenic inputs, water quality, and habitat quality than previously considered. Alternatively, perhaps their lower vagility limits them to more persistent waterbodies: even if damselflies can develop quickly enough from an egg to emerge from a temporary pool, the adults would have to disperse to a new water source when their natal sites dry up, a common occurrence in a desiccating environment.

All of the urban areas in our study were of modest size in terms of population density and spatial extent; Lubbock is the largest regional city, with a population of 258,862 within 320 km^2^ [52]. Because the definition of what is urban is not universal nor consistent, however [53,54], there may be differences in odonate responses to what we called “urban” environments. This is of particular importance, given that odonate diversity responds to a gradient of urban disturbance [23,55,56]. Urban ponds and lakes should be considered as important areas of odonate diversity conservation, especially in arid to semi-arid areas. Although it may be tempting to disregard them as human-made, and therefore not true or adequate proxies of natural areas, if properly designed with nature in mind, they can support wildlife [57]. For example, urban ponds can be designed to mimic the habitat quality of natural systems by planting emergent vegetation, the use of vegetation buffer zones to filter pollutants, and clearing overgrown vegetation to increase insolation [57,58]. Odonate breeding habitats are at least somewhat independent of surrounding terrestrial habitats, and urban wetlands can provide aquatic vegetation that would be unavailable in otherwise xeric areas. However, not all urban wetlands do this. Restoration and management of urban ponds can support the aesthetics of the public while creating favorable habitats for odonates.

Odonates capture the public eye with their array of colors, ability to fly, and their diversity worldwide. As urban areas continue to expand, let us consider viable options to continue connecting people with nature and support urban biodiversity. The creation of urban wetlands has had an unintended positive effect on odonate richness in the Texas panhandle, and could perhaps be a gateway for the potential to design habitats with such diversity in mind. Considering urban freshwater wetlands as modified aquatic habitats in lieu of natural areas may be the ultimate concession to preserve both regional odonate diversity and freshwater in this region.

## 5. Conclusions

The Texas panhandle has been under-sampled for odonates, as most of the ~20,000 playas occur on private property. Restricted access, compounded by short hydroperiods and lengthy periods of time when water is entirely absent, limit our understanding of species presence in playas of this region. Using a long-term dataset (2003–2020), we found that urban playas had longer hydroperiods than did playas surrounded by cropland or grassland land use/land cover types, and harbored odonate communities with greater species richness and a different composition than playas in other croplands or grasslands. In this semi-arid region, urban-mediated, increased hydroperiods may play an important role in supporting greater odonate richness, particularly in times of water scarcity. These findings indicate that anthropogenic activities create refugia for odonates in a semi-arid area.

## Figures and Tables

**Figure 1 insects-12-00431-f001:**
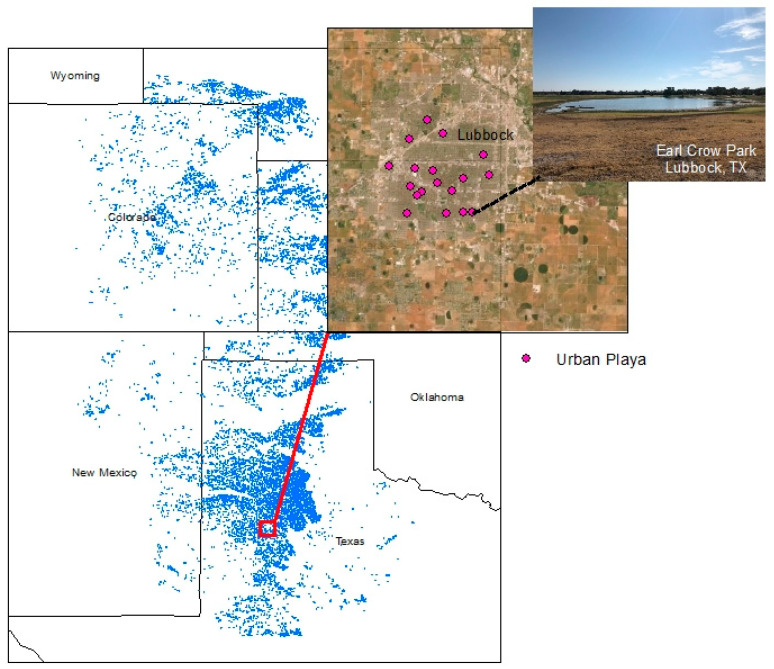
Map of playas of the southern Great Plains, with insets of urban playas sampled in the city of Lubbock, Texas (pink dots), and a photograph of one such example (Earl Crow Park). Basemap from Esri, playa data from Playa Lakes Joint Venture: https://pljv.org/for-habitat-partners/maps-and-data/data-downloads/ (accessed on 17 February 2021); inset map from Esri, photograph by D.M. Husband.

**Figure 2 insects-12-00431-f002:**
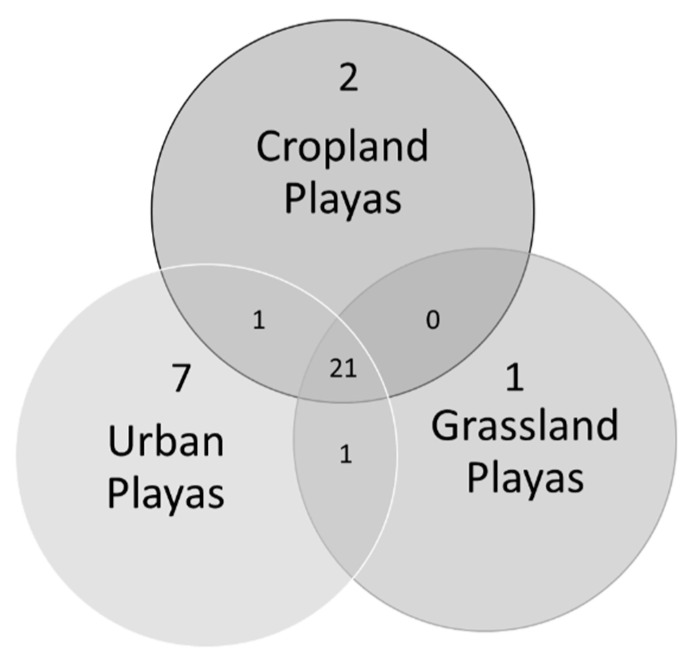
Venn diagram of odonate species occurrences at playa wetlands in the Texas panhandle (United States), 2003–2020, characterized by the dominant land use/land cover type within 200 m. Grassland, cropland, and urban playas had 21 odonate species in common.

**Figure 3 insects-12-00431-f003:**
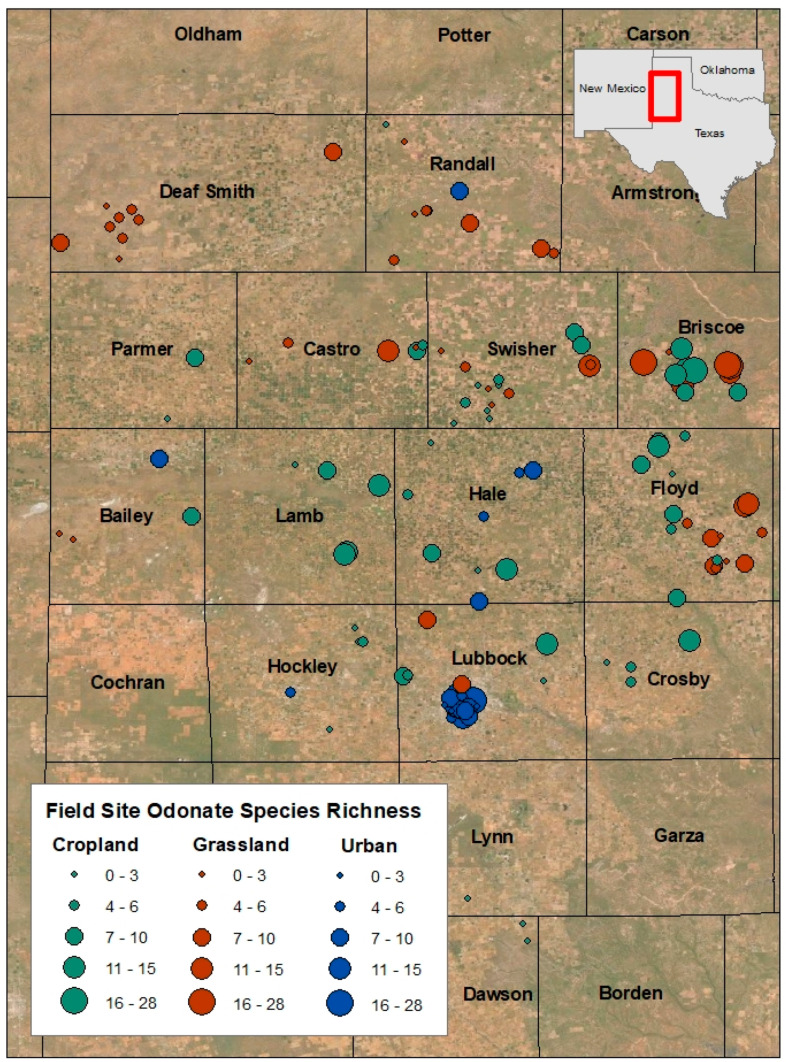
Map of field study sites by surrounding land use/land cover type, with symbols sized commensurately to odonate species richness across the Texas panhandle (United States), 2003–2020; categories were based on natural breaks in the data. Basemap layer from Esri (Redlands, CA, United States).

**Figure 4 insects-12-00431-f004:**
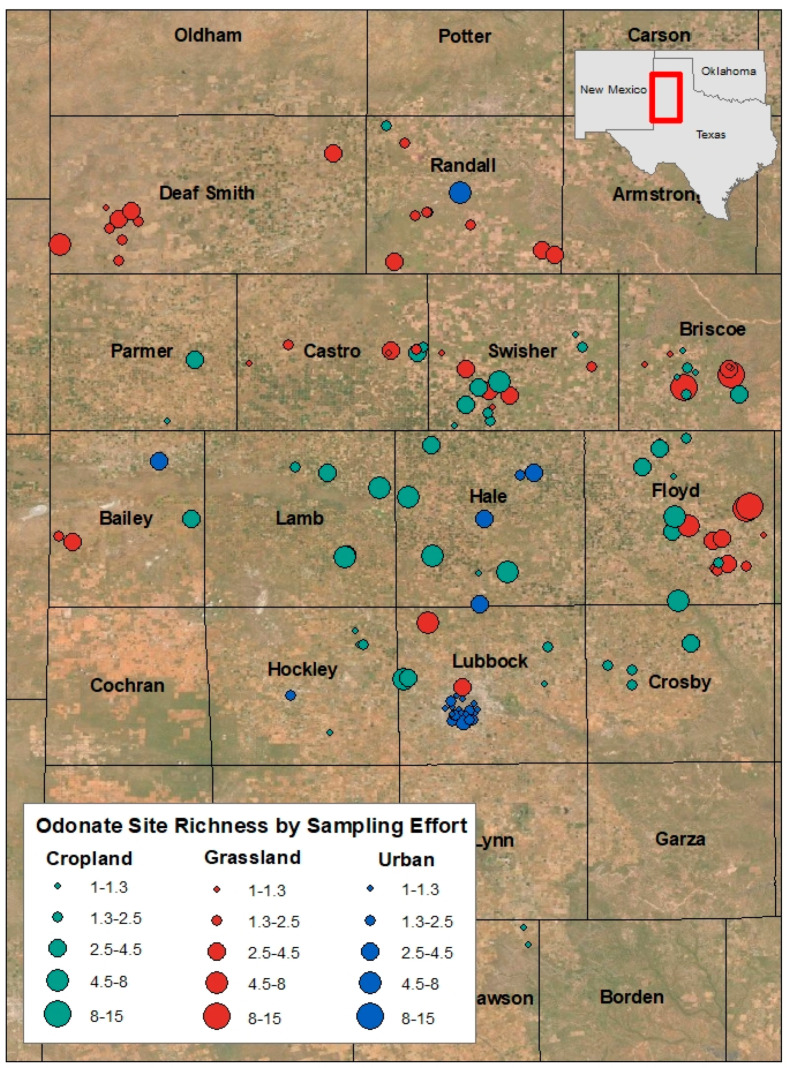
Map of field study sites by surrounding land use/land cover type, with symbols sized commensurately to odonate species richness, scaled by sampling effort (number of sampling visits) across the Texas panhandle (United States), 2003–2020; categories were based on natural breaks in the data. Basemap layer from Esri (Redlands, CA, United States).

**Figure 5 insects-12-00431-f005:**
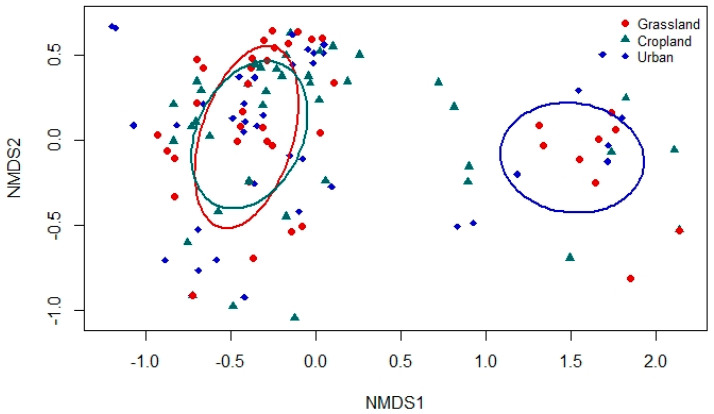
Non-metric, multidimensional scaling ordination plot with 95% confidence ellipses of the centroids in an ordination space of odonate species occurrences at three types of playa environments (red circles = grassland; green triangles = cropland; blue diamonds = urban) across the Texas panhandle (United States), 2003–2020.

**Figure 6 insects-12-00431-f006:**
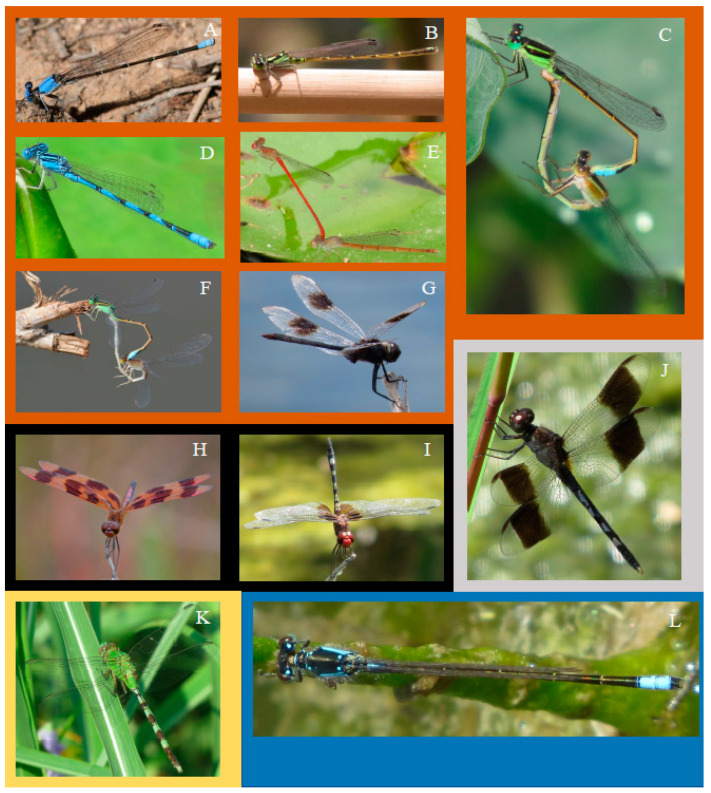
Species that were not found at all three playa wetland types across the Texas panhandle (United States), 2003–2020. Orange: urban playa species ((**A**) *Argia apicalis*, *(***B**) *Ischnura posita*, (**C**) *Ischnura ramburii*, (**D**) *Enallagma basidens*, (**E**) *Telebasis salva*, (**F**) *Ischnura barberi*, (**G**) *Brachymesia gravida*). Black: cropland playa species ((**H**) *Celithemis eponina*, (**I**) *Dythemis fugax*. Gray: grassland playa species ((**J**) *Erythrodiplax umbrata*). Yellow: urban and grassland playa species ((**K**) *Erythemis vesiculosa*). Blue: cropland and urban playa species ((**L**) *Ischnura damula*). Photographs by N.E. McIntyre.

**Figure 7 insects-12-00431-f007:**
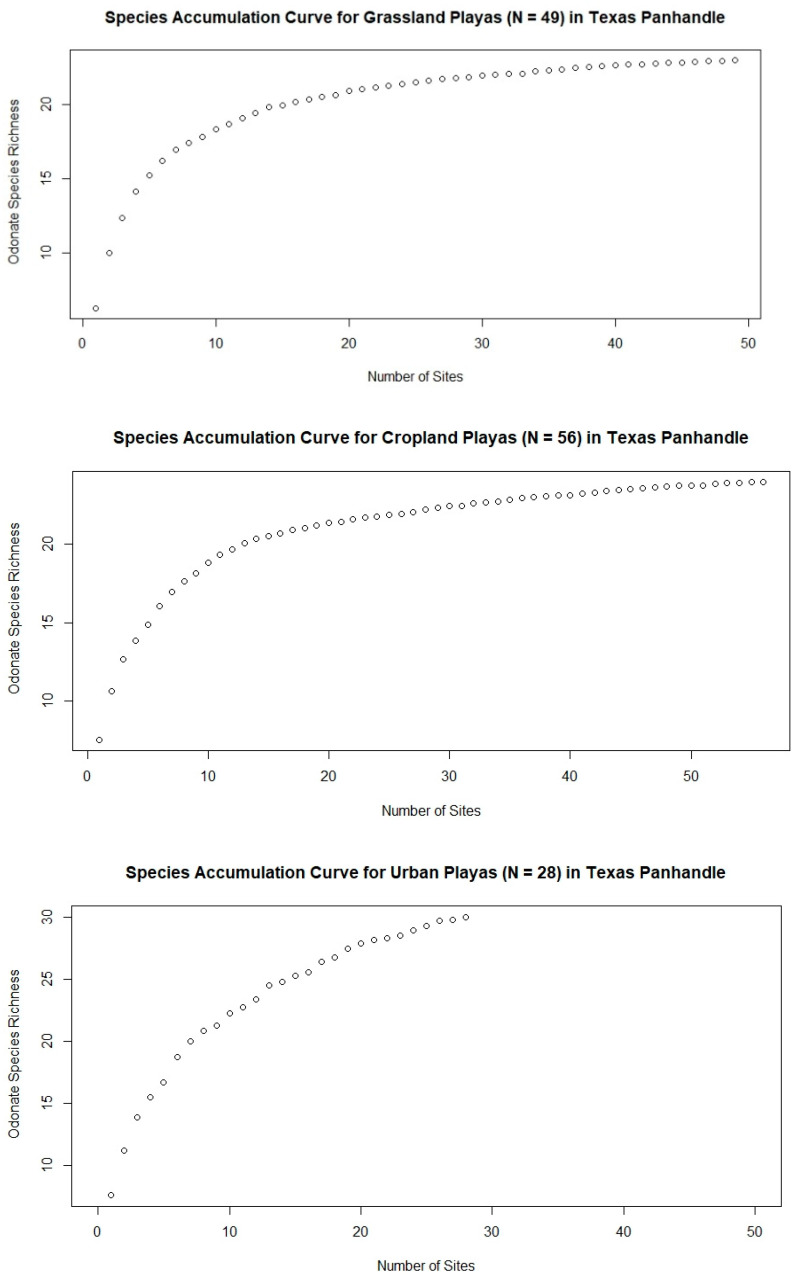
Species accumulation curves by surrounding land use/land cover.

**Table 1 insects-12-00431-t001:** Odonate species occurrences (“1” indicates presence) listed taxonomically in playa wetlands characterized by the dominant land use/land cover type within 200 m, Texas panhandle (United States), 2003–2020.

Species	Grassland	Cropland	Urban
	(*n* = 49)	(*n* = 56)	(*n* = 28)
Zygoptera			
*Lestes alacer*	1	1	1
*Lestes australis*	1	1	1
*Enallagma civile*	1	1	1
*Enallagma basidens*			1
*Ischnura ramburii*			1
*Ischnura barberi*			1
*Ischnura damula*		1	1
*Ischnura demorsa*	1	1	1
*Ischnura denticollis*	1	1	1
*Ischnura posita*			1
*Ischnura hastata*	1	1	1
*Telebasis salva*			1
*Argia apicalis*			1
Anisoptera			
*Rhionaeschna multicolor*	1	1	1
*Anax junius*	1	1	1
*Plathemis lydia*	1	1	1
*Libellula saturata*	1	1	1
*Libellula pulchella*	1	1	1
*Libellula luctuosa*	1	1	1
*Orthemis ferruginea*	1	1	1
*Perithemis tenera*	1	1	1
*Brachymesia gravida*			1
*Celithemis eponina*		1	
*Erythemis vesiculosa*	1		1
*Erythemis simplicicollis*	1	1	1
*Erythrodiplax umbrata*	1		
*Sympetrum corruptum*	1	1	1
*Pachydiplax longipennis*	1	1	1
*Dythemis fugax*		1	
*Tramea onusta*	1	1	1
*Tramea lacerata*	1	1	1
*Pantala flavescens*	1	1	1
*Pantala hymenaea*	1	1	1

**Table 2 insects-12-00431-t002:** Frequency of the occurrence of water in playas sampled in the Texas panhandle (United States), 2003–2020.

Playa Type (*n*)	% Wet	% Dry	Total Site Visits
Grassland (51)	52	48	88
Cropland (56)	53	47	49
Urban (28)	98	2	28

## Data Availability

The data presented in this study are available on request from the corresponding author. The data are not publicly available due to ongoing analyses as part of D.M.H.’s Master’s thesis research.

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
