# Peer review of "Urban Areas Create Refugia for Odonates in a Semi-Arid Region"

_insects, 2021, doi:10.3390/insects12050431_

Round 1

Reviewer 1 Report

I think it would be worthwhile to devote a few sentences to the idea that odonate breeding habitats (i.e., freshwater wetlands of all sorts) are at least somewhat independent of surrounding terrestrial habitats. In my state, an aquatic habitat can occur on the wet Olympic Peninsula surrounded by conifer forest (as long as the conifers don't shade the wetland) and in the dry Columbia Basin surrounded by sagebrush steppe and can have virtually the same list of odonates. I think this is one of the factors that make urban wetlands valuable to them, as long as those wetlands have a complement of aquatic vegetation.

Indeed, forests are important to odonates--but not to all of them.

p, 5, middle of second paragraph - inconsistencies in use of generic names

p. 6, 9 Something happened with Figs. 1 and 4 in the copy I reviewed, with the figure separating the explanatory text. Presumably this won't be a problem in the finished paper.

p. 8, 9 - Figures 3 & 4 are superb, but unfortunately, they don't impress the eye with the greater value of urban playas to odonates, as there seem to be as many or more large red and green as blue circles. This may be due to the range of species in the largest circles, and if there had been more division at the high end, e.g.,16-20-species and 21-28-species circles, perhaps the differences would have been more striking. Or perhaps the scale is the problem, with the biodiverse urban playas around Lubbock unable to be expressed well in the figures. But I can see that it might be too much effort to modify this.

p.9, Figure 4 - Note the caption says that the green symbols are triangles and the blue symbols diamonds, which is mostly not the case.

p. 12, Table 1 - This is of course a finished study and is fine the way it is, but future studies should consider at least an effort to indicate degree of use. Surveys could rank species as "one," "few" or "many" adults seen or some metric of relative abundance. Might be worth emphasizing this.

p. 13 - Figure 5 is referenced, but this was lacking from my review copy. It shouldn't matter.

Author Response

Please see the attachment (PDF).

Reviewer 2 Report

The manuscript entitled “Anthropogenic Activities Create Refugia for Odonates in a Semi-arid Area” by Husband and McIntyre is an interesting study analyzing richness of Odonata in playas of western Texas. It shows how diversity of Odonata is higher in urban playas than others. The work deserves to be published, but in my opinion some points should be resolved before publication. Hereafter, some comments and suggestions for authors.

The term anthropogenic designates something derived from human activity, for this reason also croplands are considered anthropogenic. Authors have found a difference between urban Playas communities in respect to cropland and grassland one, this means that “Urban areas” Create Refugia for Odonates in a Semi-arid Area and actually not Anthropogenic Activities per se. In light of this, I suggest revising the manuscript title and conclusive statements.

Materials and methods

Authors stated they identified only adults and not pre imaginal stages since “identifying exuviae or nymphs (particularly of early instars and of Zygoptera) to species in this area is difficult”. Why doing it in this area is more difficult than in others? Please specify it in the sentence.

In the paragraph “Odonate Assemblage Characterization and Analysis” authors present NMDS analysis. In my opinion some unnecessary details on this analysis are reported, e.g., explanation of what NMDS is, specifying how they set the arguments of R function “(with a binary = TRUE statement to denote presence/absence data)” .

Results

Authors never report the list of species that were identified associated with information on occurrence locality (including coordinates) and date of recovery. In my opinion this is fundamental information for mapping each species record in the area. These data should be presented, maybe also as supplementary file.

Moreover, in my opinion, too many figures and tables are present in this section, some of them superfluous. Hereafter, I report some suggestion to authors regarding this point.

Table 2 could be removed since the information reported is redundant, the same information can be derived summing columns of Table 1.

The details on how values reported in Table 3 were calculated is not clear to me. In materials and methods there is a sentence explaining how extent of water was recorded, but I think these values are not related to those records, since the former were registered only for 82 playas. In this case, what “total sampled” is? Areas are 133 in total but “total sampled” sum is 165.  In addition, why percentages of wet and dry do not sum to 100?

In my opinion, a table could be devoted to ANOVA results.

Figure 3 and 4 could become a single figure, i.e. Fig. 3 A and B.

Regarding figure 7, species accumulation curves could be resumed in a single graph; confidence intervals should be added.

Finally, in my view figures legend should not report sentences of discussion or results, e.g. Figure 7 “The species accumulation curve based on sampling at 28 urban playas did not reach an asymptote, indicating that odonate richness at urban playas is likely under-estimated”. Similarly, Figure 2 “Grassland, cropland and urban playas had 21 odonate species in common. Seven species were solely found at urban playas. Cropland and grassland playas shared no species not also found at urban playas. Grassland and urban playas shared one species not found at cropland playas. Urban and cropland playas shared one species not found at grassland playas.”

Author Response

Please see the attachment (PDF).
